May 31, 2024

# Retraction Notice

Retraction: Guo X, Zhang J, Zeng J, Guo Y, Zhao L. 2023. MiR-525-5p inhibits diffuse large B cell lymphoma progression via the Myd88/NF-κB signaling pathway. PeerJ 11:e16388 https://doi.org/10.7717/peerj.16388

After concerns were raised by a reader, an investigation by PeerJ has confirmed that:

A) the U2932 cell line reportedly used in this work has a small and round morphology, with cells growing singly and in clusters in suspension. However, photographs in figs. 4C and 5C show typical epithelial cell morphology suggesting the use of a contaminated cell line.

B) The protocols detailed in the Methods section for the Transwell assays do not include any modification to ensure that U2932 cells (which are a suspended cell line) would not be washed away when removing the culture medium. Therefore, even if U2932 cells had been present, they could not have been detected by this method.

The authors have been made aware of these issues, and confirmed that the maintenance conditions of their cells were conducive to inadvertent contamination. In agreement with the authors, the Publisher has therefore taken the decision to retract this article.

PeerJ Editorial Office. 2024. Retraction: MiR-525-5p inhibits diffuse large B cell lymphoma progression via the Myd88 /NF-κB signaling pathway. PeerJ 12:e16388/retraction
https://doi.org/10.7717/peerj.16388/retraction.

PeerJ Editorial Office. 2024. Retraction: MiR-525-5p inhibits diffuse large B cell lymphoma progression via the Myd88 /NF-κB signaling pathway. PeerJ 12:e16388/retraction
https://doi.org/10.7717/peerj.16388/retraction



# MiR-525-5p inhibits diffuse large B cell lymphoma progression via the Myd88/NF-κB signaling pathway

Xiuchen Guo[1], Jingbo Zhang[1], Jingya Zeng[2], Yiwei Guo[1] and Lina Zhao[1]

[1] Department of Hematology, Harbin Medical University Cancer Hospital, Harbin, China
[2] Department of Clinical Laboratory, Harbin Medical University Cancer Hospital, Harbin, China

## ABSTRACT

Diffuse large B-cell lymphoma (DLBCL) is a B-cell lymphoma with a high degree of aggressiveness. Recently, evidence has shown that miR-525-5p is decreased in DLBCL, suggesting its possible involvement in tumor progression. In this study, miR-525-5p suppressed proliferation, invasion and clonogenicity, and increased apoptosis of U2932 cells, whereas miR-525-5p silencing enhanced tumor cell growth. Next, miR-525-5p targets the 3′-UTR of Myd88, and Myd88 protein was increased in lymphoma tissues. Similar to the miR-525-5p mimic, Myd88 siRNA suppressed proliferation, invasion, and clonogenicity, and enhanced apoptosis of U2932 cells. We observed that Myd88 reversed the inhibitory effect of miR-525-5p on tumor cell growth by transfecting cells with miR-525-5p mimics alone or together with Myd88 overexpression vector. In addition, *in vivo* studies have shown that compared to the control group, U2932 cells with upregulated miR-525-5p expression have a reduced ability to induce tumor formation. In conclusion, our results demonstrate that miR-525-5p inhibits the progression of DLBCL through the Myd88/NF-κB pathway, which largely fills the gap of previous studies, and our results may provide a new reference for the targeted treatment of DLBCL.

## INTRODUCTION

Diffuse large B-cell lymphoma (DLBCL) is the most common clinical subtype among non-Hodgkin lymphomas, and its incidence is statistically estimated to account for more than 40% of aggressive lymphomas (*Liu et al., 2022*). DLBCL is highly aggressive and heterogeneous, and therapeutically, although high clinical remission rates and survival rates have been achieved, still greater than 30% of DLBCL patients are refractory or relapse after first-line standard treatment (*Liu et al., 2019*). With the continuous innovation of molecular biology techniques, the distinction of each DLBCL subtype has been revealed from the genetic level (*de Leval et al., 2022*; *Riedell & Smith, 2018*). Currently, more in-depth studies are needed to explore the pathogenesis of DLBCL before new therapeutic targets can be found, individualized therapy for patients is proposed, and long-term patient outcomes are improved.

Corresponding author
Lina Zhao, 1601@hrbmu.edu.cn

Recently, more and more studies have shown that DLBCL occurrence and development are closely related to microRNAs (miRNAs). MiRNAs mainly lead to mRNA degradation or translational repression by binding to the 3′-UTR of target gene mRNAs and play epigenetic modifying roles in cell differentiation, development, proliferation and apoptosis (*Solé et al., 2017*; *Drees & Pegtel, 2020*). Altered expression of miRNAs is found in almost all types of tumors, and miRNA regulated cell signaling pathways exert oncogenic or tumor suppressor effects and participate in regulating tumor proliferation, invasion, metastasis, and therapeutic resistance, affecting tumor progression. MiR-525-5p has been reported to play a tumor suppressive role in most different kinds of tumors. In cervical cancer progression, miR-525-5p inhibited cell proliferation, invasion and migration (*Chen & Liu, 2020*). Another study showed that miR-525-5p attenuated glioma cell viability, and inhibited migration and epithelial mesenchymal transition (*Xie et al., 2020*). Furthermore, in ovarian, colorectal, and breast cancer progression, miR-525-5p inhibited proliferation and invasion of tumor cell, and increased cell apoptosis (*Wang et al., 2020*; *Liu et al., 2020*; *Xiao et al., 2021*). Results from an analysis of 377 miRNAs in 32 Hodgkin lymphoma patients demonstrated that miR-525-3p was downregulated in patients with tumors compared with samples from patients with reactive lymphadenopathy, suggesting that differential expression of miR-525-3p is involved in lymphoma progression (*Paydas et al., 2016*).

Myd88 is involved in mediating innate immune responses by receiving signals efferent from upstream TLRs and integrating the transduction. As a signal transduction hub, Myd88 participates in multipathway signal transduction and participates in tumor initiation and progression through multiple mechanisms. In the progression of lung cancer, inhibition of Myd88 pathway could reduce the expression of PD-L1, thereby inhibiting the growth of A549 cells and inducing apoptosis (*Zhao et al., 2023*). In addition, a study showed that Myd88 signaling in myofibroblasts increased the secretion of CCL20, which promoted aerobic glycolysis of hepatocellular carcinoma cells and accelerated tumor cell growth (*Yuan et al., 2022*). *Zhu et al. (2019)* found that Myd88 signaling plays a key role in pancreatic cancer-induced inflammation, thereby triggering the development of cachexia. It has been shown that aberrant activation of these signaling pathways is closely associated with DLBCL pathogenesis, malignant proliferation, invasion, metastasis, drug resistance and poor prognosis. Some scholars have found that there is a higher frequency of Myd88 mutations in DLBCL, which is the main reason for triggering abnormal activity of the nuclear factor kappa B (NF-κB) pathway (*Yu et al., 2018*). However, although there are a few studies showing high Myd88 expression in DLBCL, the specific mechanism of action is not fully understood.

The role of miR-525-5p in the progression of DLBCL has not been reported in the existing studies. Therefore, in order to fill the gaps in previous studies, in this study, we observed the proliferation, invasion and apoptosis of DLBCL cell lines by overexpressing or knocking down the expression of miR-525-5p and Myd88, and constructed transplanted tumor nude mice overexpressing miR-525-5p to verify the regulatory role of miR-525-5p and Myd88 in the progression of DLBCL, in order to provide new ideas for the treatment of DLBCL.
## MATERIALS AND METHODS

### Cell culture and clinical samples

Human lymphoblastoid B cells (GM12878), and lymphoma cell lines OCl-LY7, FARAGE and U2932 were purchased from ATCC. The cells were incubated in DMEM (10% FBS, 100 U/mL penicillin, 100 µg/mL streptomycin). The miR-525-5p mimic, miR-525-5p inhibitor and negative control were purchased from Santa Cruz Biotechnology, Inc. (Dallas, TX, USA). The Myd88 overexpression vector (OE-Myd88) and vector were purchased from RiboBio. Myd88 siRNA (F: CCG GGC CTA TCG CTG TTC TTG AAT TCA AGA GAT TCA AGA ACA GCG ATA GGC TTT TTT GGT ACC; R: AAT TGG TAC CAA AAA AGC CTA TCG CTG TTC TTG AAT CTC TTG AAT TCA AGA ACA GCG ATA GGC) and scramble were purchased from Santa Cruz Biotechnology, Inc. A total of 20 pairs of tumor tissues and paracancerous tissues were obtained from patients. The area more than 5 cm away from the edge of tumor tissue was taken as paracancerous tissue, and tumor tissue samples were processed as follows: the tumor tissues were washed with normal saline to remove blood vessels, envelopes, and necrotic tissues, and the tumor tissues were cut into 1-mm-thick slices, and next, the tumor tissues were spread on the scaffolds after soaking in vitrification solution and quickly put into sterile liquid nitrogen for storage. All samples obtained in this study were approved by the ethics committee of the Harbin Medical University Cancer Hospital and abided by the ethical guidelines of the Declaration of Helsinki, and ethics committee agreed to waive informed consent (Approval number: XJS2022-18).

### Animal

U2932 cells transfected with NC mimic or miR-525-5p mimic, respectively, were injected subcutaneously into nude mice to obtain tumor xenografts. A total of 10 adult nude mice (4–6 weeks, 18,722 g; Wuhan Experimental Animal Center, Wuhan, China) were randomly divided into two groups ($N = 5$ per group) including NC mimic group and miR-525-5p mimic group. The mice were housed in a clean and well-ventilated animal environment at $20 \pm 2$ °C, with a relative humidity of 60–70% and a day/night cycle of 12/12 h. They had free access to water and food. On the 35th day after injection, we collected tumor tissues. All protocols on animals conformed to the laboratory animal guidelines. This study was approved by the Animal Ethics Committee of Harbin Medical University Cancer Hospital.

### CCK-8 assay

U2932 cells were seeded at $2 \times 10^4$ cells/well in 96 well plates, and 0, 24, 48, and 72 h after transfection, the cell culture supernatant was discarded. Subsequently, 20 µL of CCK-8 solution (Dojindo, Mashiki, Japan) was added to each well, and incubated in a 37 °C incubator for 4 h, and the absorbance at 450 nm was measured by a microplate reader (Thermo Fisher Scientific, Waltham, MA, USA).

## Flow cytometry

Firstly, the cells were digested with trypsin and the culture medium was rinsed with PBS to obtain a cell suspension. Then, 100 µL of cell suspension was transferred into the culture tube, and V-FITC and PI was added to the cells. Next, the cells were incubated in darkness for 20 min, and flow cytometry (BD Biosciences, Franklin Lakes, NJ, USA) was used to detect cell apoptosis. Among them, Annexin V-FITC single positive cells were early apoptotic cells, and cells double positive for Annexin V-FITC and PI staining were necrotic cells or late apoptotic cells.

## Western blotting

The protein was extracted by RIPA lysis buffer, and the protein content of each sample was determined using the BCA Protein Assay Kit (Thermo Fisher Scientific, Waltham, MA, USA). Then, equal amounts of proteins (15 µg/lane) were separated on a 12% sodium dodecyl sulfate polyacrylamide gel electrophoresis (SDS-PAGE) and protein was transferred to PVDF membranes (Bio-Rad, Hercules, CA, USA). After blocking, the membranes were incubated with primary antibodies (Abcam, Cambridge, UK) for 12 h. Mouse monoclonal anti-GAPDH antibody (1:3,000), rabbit monoclonal anti-Myd88 antibody (1:2,000) and rabbit polyclonal anti-NF-κB antibody (1:4,000). Then, the membranes were incubated with HRP-conjugated goat anti-rabbit IgG for 1 h at room temperature. The protein bands were visualized with ECL detection reagents and analyzed with ImageJ software (National Institutes of Health, Bethesda, MD, USA).

## RT-qPCR

Total RNA was isolated by the TRIzol, and Real-time qPCR was conducted by using a SYBR Premix Ex TaqTM Kit (Applied Biosystems, Waltham, MA, USA). The primers (Sangon, Shanghai, China) are as follows: miR-525-5p forward, 5′-GCG GTC CCT CTC CAA ATG T-3′, reverse, 5′-AGT GCA GGG TCC GAG GTA TT-3′; U6 forward, 5′-CTC GCT TCG GCA GCA CA-3′, reverse, 5′-AAC GCT TCA CGA ATT TGC GT-3′; Myd88 forward, 5′-GGC TGC TCT CAA CAT GCG A-3′, reverse, 5′-CTG TGT CCG CAC GTT CAA GA-3′; GAPDH forward, 5′-GAA GGT GAA GGT CGG AGT C-3′, reverse, 5′-GAA GAT GGT GAT GGG ATT TC-3′. The relative expression levels were normalized by using the $2^{-\Delta\Delta Ct}$ method.

## Luciferase reporter gene assay

First, the wild-type and mutant Myd88 3′-UTR sequences were fused into the luciferase reporter vector respectively, and then the cells were transfected with vector and miR-525-5p mimic or NC mimic. After 48 h of cultivation, the luciferase activity was measured using an enzyme-linked immunosorbent assay.

## Transwell cell invasion assay

Cells were inoculated into the upper cavity of Transwell coated with matrix, and culture medium was added into the lower cavity of Transwell. The cells were cultured in an incubator at 37 °C for 48 h, the upper lumen cells were wiped with a cotton swab, the lower

lumen cells were fixed with 70% ethanol, and the lower lumen cells were stained with crystal violet, and the invasion of cells was observed under the light microscope.

## Cell colony formation assay

Firstly, the cells were inoculated into a six-well plate and supplemented with new culture medium every 72 h. After 14 days of cultivation, the cell culture supernatant was discarded using a pipette and the cells were fixed with paraformaldehyde. Then, the cell proliferation community was subjected to crystal violet staining and the proliferating cells were counted.

## Statistical analysis

The SPSS software was used to analyze all dates (ver. 18.0; SPSS, Inc., Chicago, IL, USA). The quantitative data derived from three independent experiments were expressed as mean ± SD. The Shapiro-Wilk test was utilized to verify the data's normal distribution, while Levene's test was employed to assess the homogeneity of variance. $P < 0.05$ means statistically significant.

# RESULTS

## MiR-525-5p expression in lymphoma tissues and cells

We collected 20 pairs of tumor tissues and paracancerous tissues from lymphoma patients, and RT-qPCR results showed that miR-525-5p was downregulated in tumor tissues (Fig. 1A), and compared with healthy subjects, the miR-525-5p expression was also downregulated in peripheral blood of lymphoma patients (Fig. 1B). Compared with human lymphoblastoid B cells (GM12878), miR-525-5p was downregulated in the lymphoma cell lines OCl-LY7, FARAGE and U2932 (Fig. 1C). In addition, the U2932 cells were transfected with miR-525-5p mimic or miR-525-5p inhibitor, and we found that miR-525-5p mimic promoted miR-525-5p mimic expression, and miR-525-5p inhibitor inhibited miR-252-5p expression (Fig. 1D).

## MiR-525-5p inhibited U2932 cell growth

To explore the effect of miR-525-5p on the growth of U2932 cells, the U2932 cells were transfected with miR-525-5p mimic or miR-525-5p inhibitor, respectively. We found that miR-252-5p mimic inhibited cell proliferation (Fig. 2A), promoted cell apoptosis (Fig. 2B), and attenuated cell invasion (Fig. 2C) and clonogenicity (Fig. 2D), and the silencing of miR-525-5p with miR-525-5p inhibitor promoted these malignant features of U2932 cell on cell growth.

## MiR-525-5p targets the 3′-UTR of Myd88

Through online bioinformatics databases (Starbase 3.0; http://www.sysu.edu.cn/), we found that Myd88 is the target mRNA for miR-525-5p. Considering that Myd88 plays a cancer promoting role in various types of tumors, and it has also been reported in lymphatic cancer (*Shiratori, Itoh & Tohda, 2017*). Therefore, Myd88 was investigated as a downstream target of miR-525-5p in this study. The wild type and the mutant sequences of Myd88 3′-UTR were shown in Fig. 3A. Next, the miR-525-5p mimic weakened the luciferase activity of wild type Myd88, while the luciferase activity of mutant type Myd88

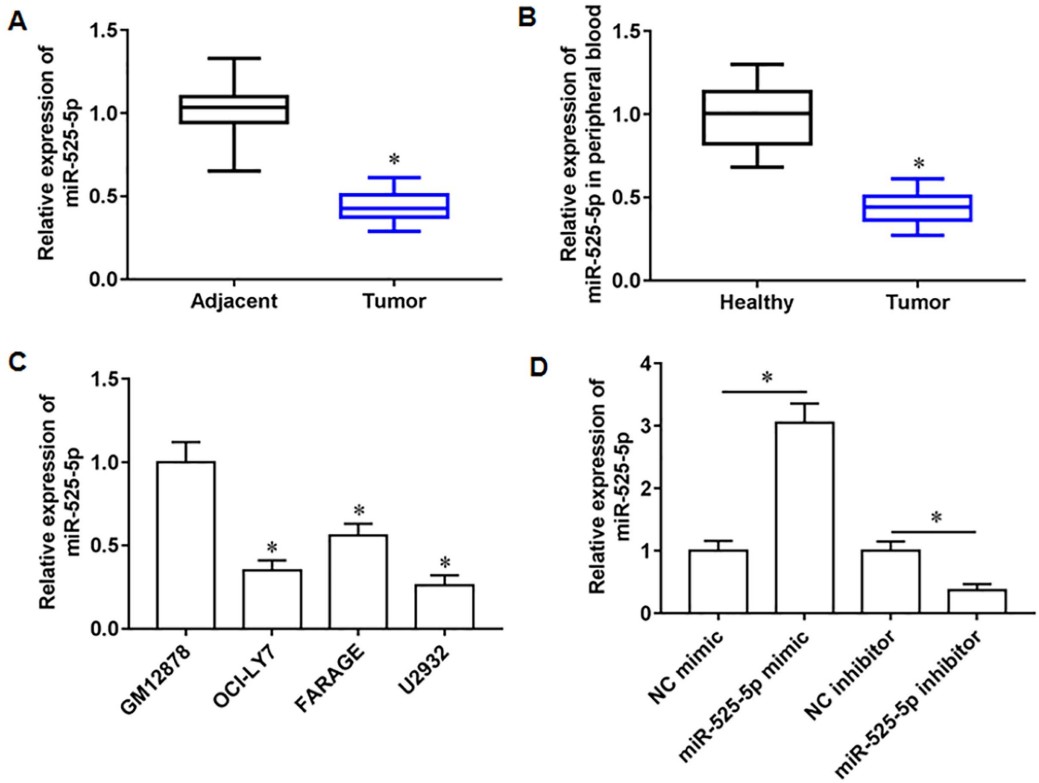

**Figure 1 The expression level of miR-525-5p in lymphoma tissues and cells.** (A) MiR-525-5p expression in tissues was detected by RT-qPCR (20 pairs of tumor tissues and paracancerous tissues). (B) MiR-525-5p expression in peripheral blood of healthy subjects ($N = 20$) and lymphoma patients ($N = 20$) was detected by RT-qPCR. (C) RT-qPCR was used to detect miR-525-5p expression in human lymphoblastoid B cells (GM12878), and lymphoma cell lines OCl-LY7, FARAGE and U2932. (D) The U2932 cells were transfected with miR-525-5p mimic or miR-525-5p inhibitor for 24 h, respectively. The transfection efficiency of miR-525-5p mimic and inhibitor was detected by RT-qPCR. $N = 5$. *$P < 0.01$.

did not change (Fig. 3B). Furthermore, miR-525-5p mimic inhibited Myd88 expression, and miR-525-5p inhibitor promoted Myd88 mRNA (Fig. 3C) and protein (Fig. 3D) expression. In addition, Myd88 protein expression was increased tumor tissues of lymphoma tissues (Fig. 3E). RT-qPCR results showed that the mRNA expression of Myd88 in lymphoid cancer cell lines OCl-LY7, FARAGE and U2932 was upregulated compared with human lymphoblastoid B cells GM12878 (Fig. 3F).

## Myd88 promotes U2932 cell growth

To explore the effect of Myd88 on the growth of U2932 cells, the U2932 cells were transfected with Myd88 overexpression vector (OE-Myd88) or Myd88 siRNA (si-Myd88), respectively. The overexpression and knockdown efficiency of Myd88 were shown in Fig. 4A. We found that overexpression of Myd88 promoted cell proliferation (Fig. 4B) and invasion (Fig. 4C), inhibited apoptosis (Fig. 4D), and enhanced cell clonogenicity (Fig. 4E), and that Myd88 silencing inhibited cell proliferation, invasion and clonogenicity, and promoted apoptosis.

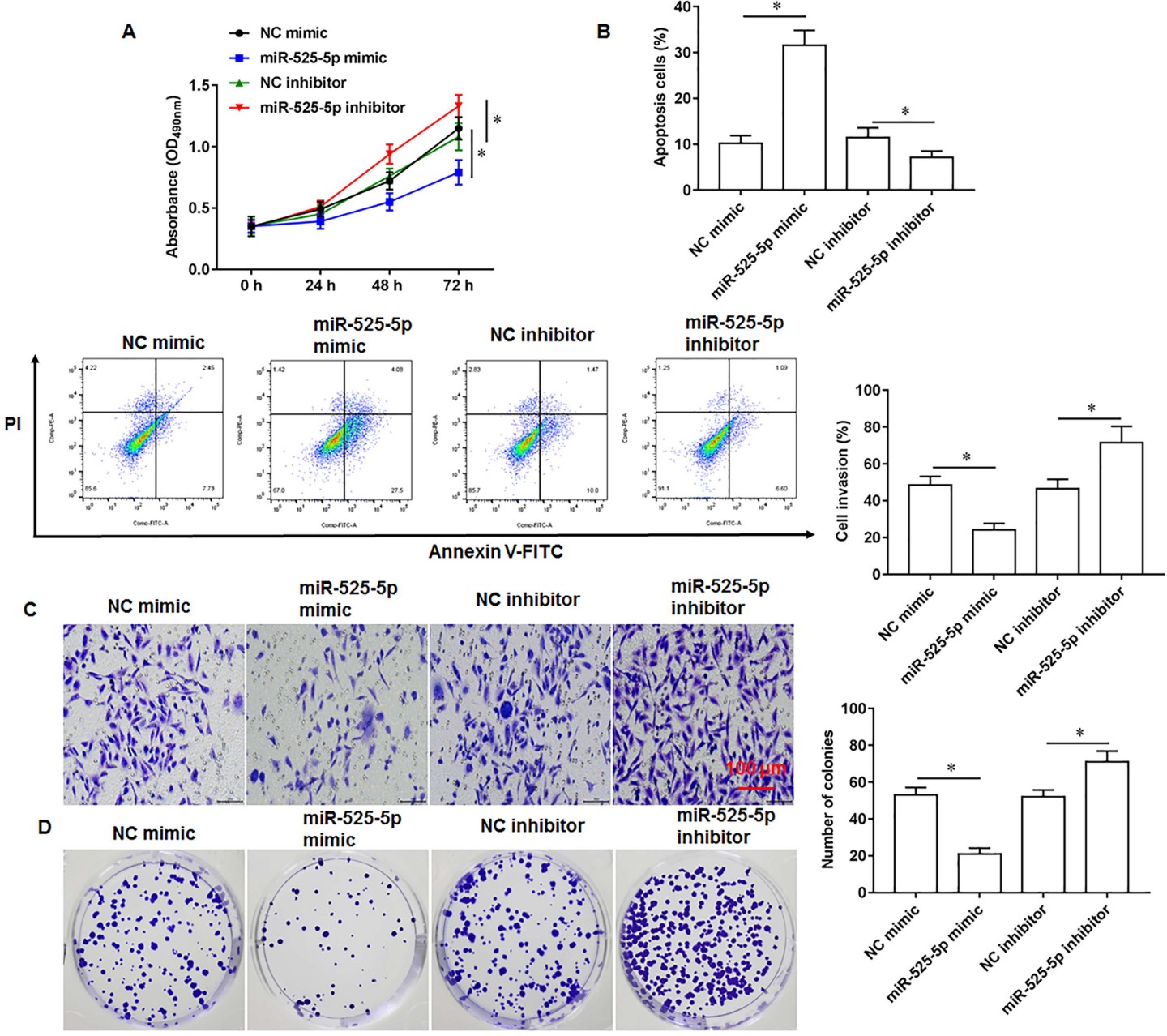

**Figure 2 Effect of miR-525-5p on lymphoma cells U2932.** The U2932 cells were transfected with miR-525-5p mimic or miR-525-5p inhibitor for 24 h, respectively. (A) CCK-8 assay was used to detect cell proliferation. (B) Cell apoptosis was detected by flow cytometry. (C) Cell invasion was detected by Transwell invasion assay. (D) Cell clonogenic capacity was detected by cell colony formation assay. $N = 5$. $*P < 0.01$.

## MiR-525-5p inhibits lymphoma cell U2932 growth by targeting Myd88

The U2932 cells were transfected with miR-525-5p mimic alone or together with Myd88 overexpression vector. We found that miR-525-5p mimic inhibited Myd88 and NF-κB protein expression, and overexpression of Myd88 again upregulated the protein expression of Myd88 and NF-κB (Fig. 5A). Furthermore, Myd88 overexpression partially abolished

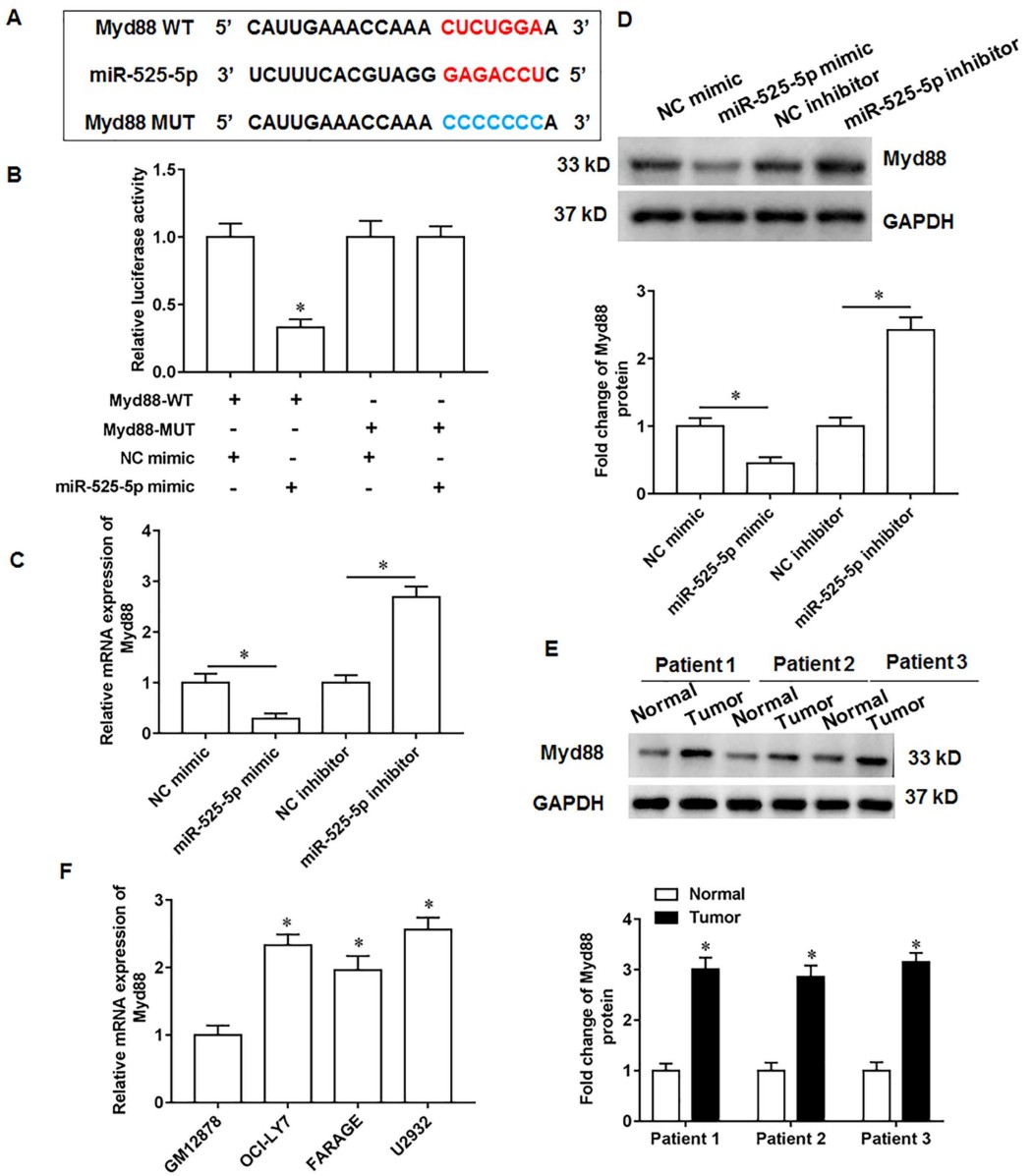

**Figure 3 MiR-525-5p targets the 3′UTR of Myd88.** (A) The wild or mutant sequences of Myd88. (B) Luciferase reporter gene analysis validate the relationship of miR-525-5p and Myd88. (C and D) Myd88 mRNA and protein expression was detected by RT-qPCR and Western blotting. (E) Myd88 protein expression. (F) The mRNA expression of Myd88 in lymphoid cancer cell lines (OCl-LY7, FARAGE and U2932) and human lymphoblastoid B cells (GM12878). $N = 5$. *$P < 0.01$.

the negative effects of miR-525-5p mimic on cell proliferation (Fig. 5B) and cell invasion (Fig. 5C), as well as the positive effects on apoptosis (Fig. 5D).

## MiR-525-5p inhibits *in vivo* tumorigenesis in nude mice

U2932 cells transfected with NC mimic (control group) or miR-525-5p mimic, respectively, were injected subcutaneously into nude mice to obtain tumor xenografts. On the 35th day after injection, we collected tumor tissues. Compared with the nude mice

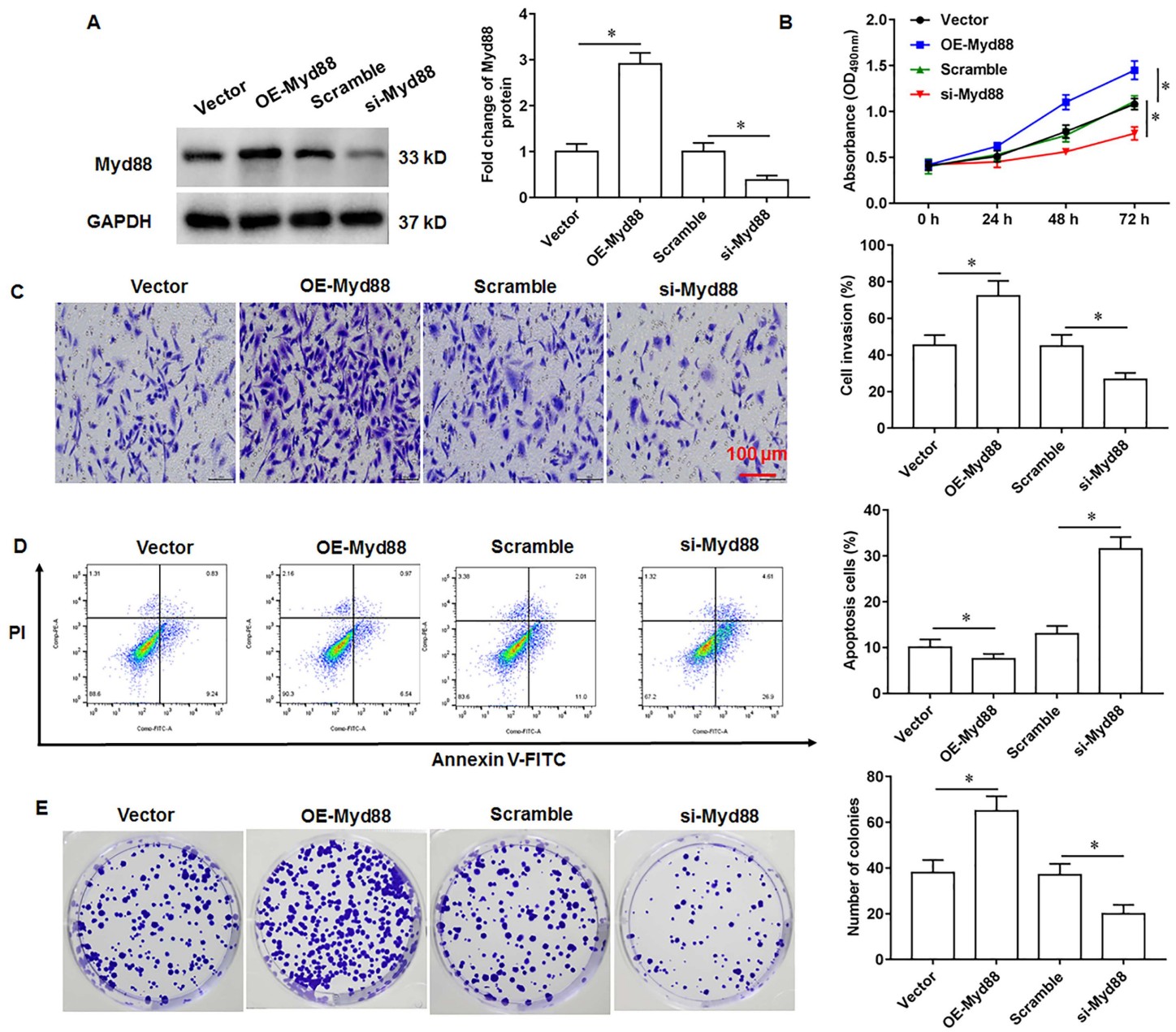

**Figure 4 Effect of miR-525-5p on lymphoma cells U2932.** The U2932 cells were transfected with Myd88 overexpression vector (OE-Myd88) or Myd88 siRNA (si-Myd88) for 24 h, respectively. (A) Myd88 protein expression was detected by Western blotting. (B) CCK-8 assay was used to detect cell proliferation. (C) Cell invasion was detected by Transwell invasion assay. (D) Cell apoptosis was detected by flow cytometry. (E) Cell clonogenic capacity was detected by cell colony formation assay. $N = 5$. $^*P < 0.01$.           

in control group, the tumor formation of nude mice overexpressing miR-525-5p was suppressed (Fig. 6A). In addition, the volume and weight of tumor tissue in the experimental group of nude mice (Fig. 6B) were smaller than those in the control group of nude mice (Fig. 6C). In addition, compared to the control group mice, the expression of miR-525-5p in the tumor tissue of the experimental group mice increased (Fig. 6D), and the protein expressions of Myd88 and NF-κB were decreased (Fig. 6E).

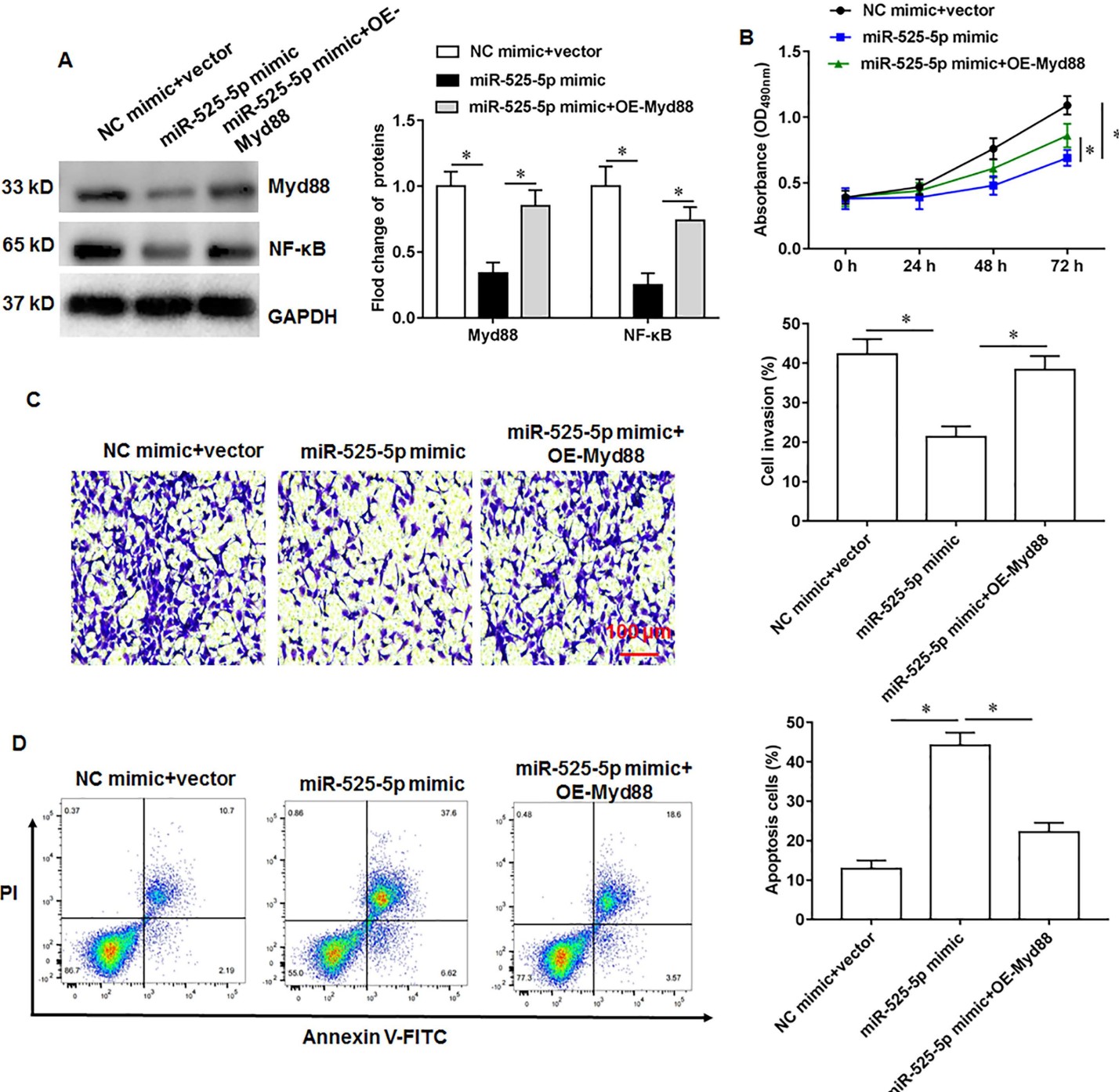

**Figure 5 MiR-525-5p affects lymphoma cells U2932 by targeting Myd88.** The U2932 cells were transfected with miR-525-5p mimic alone or together with Myd88 overexpression vector. (A) The protein expression of Myd88 and NF-κB. (B) CCK-8 assay was used to detect cell proliferation. (C) Cell invasion was detected by Transwell invasion assay. (D) Cell apoptosis was detected by flow cytometry. $N = 5$. *$P < 0.01$.

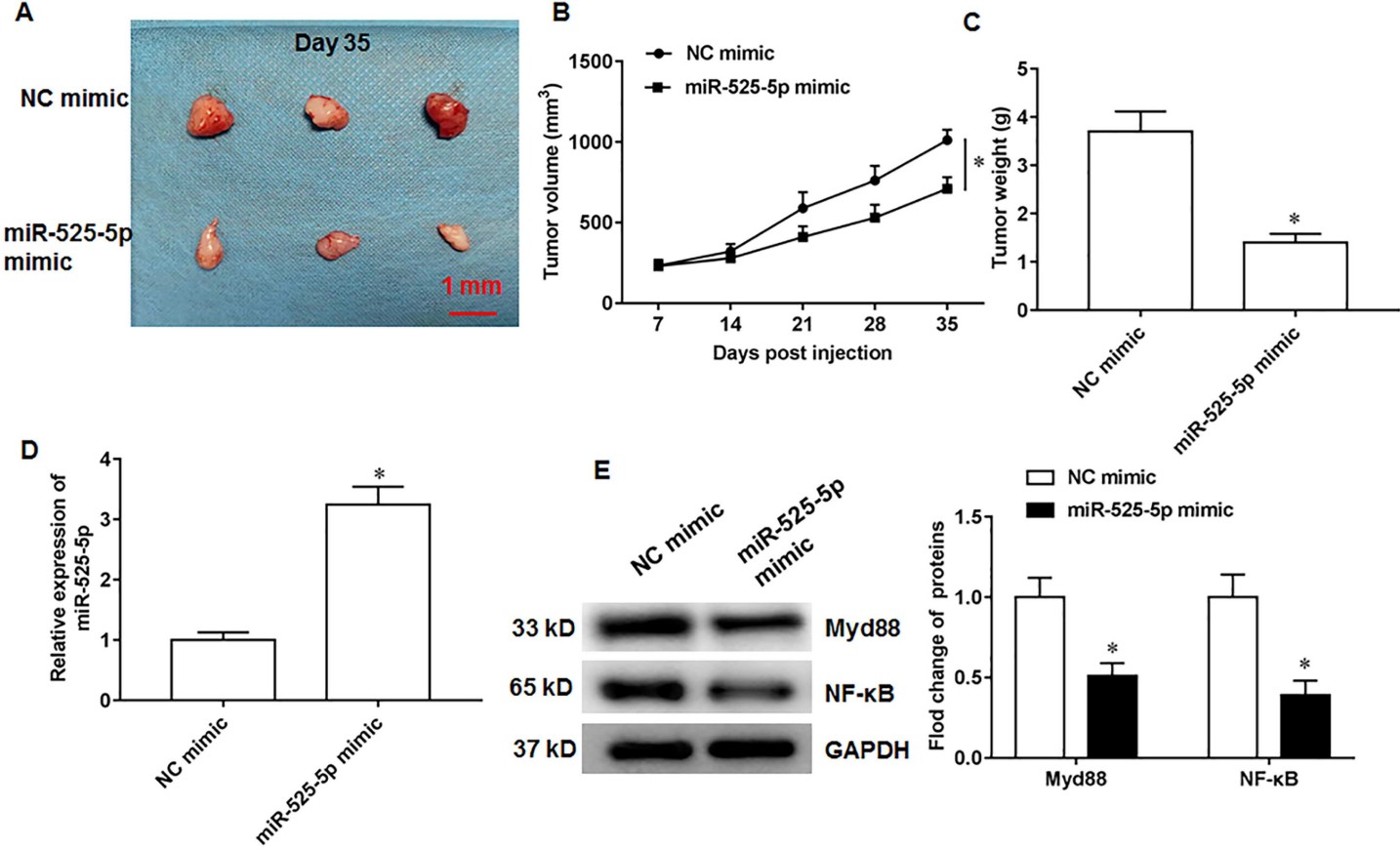

**Figure 6 MiR-525-5p inhibits tumor formation.** A total of 10 adult nude mice (4–6 weeks; 18–22 g; Wuhan Experimental Animal Center; Wuhan, China) were randomly divided into two groups ($N = 5$ per group) including NC mimic group and miR-525-5p mimic group. A total of 10 adult nude mice were randomly divided into two groups ($N = 5$ per group) including NC mimic group and miR-525-5p mimic group. (A) Tumor images on the 35th day after injection. (B) Tumor volume. (C) Tumor weight. (D) MiR-525-5p expression. (E) Myd88 and NF-κB protein expression. $N = 5$. *$P < 0.01$.

## DISCUSSION

The mechanism of DLBCL occurrence and development is very complex, involving multiple oncogenes, tumor suppressor gene mutations and complex abnormal signal transduction. In recent years, with the study of the epigenetic regulatory role of miRNAs continues to deeper, it is found that multiple miRNAs are involved in the regulation of multiple signaling pathways in DLBCL, regulating the occurrence and development of DLBCL.

MiR-525 is a small molecule miRNA encoded by chromosome 19 and comprises miR-525-5p and miR-525-3p. Currently, there are few studies on miR-525-5p in cancer, and the mechanism and functional effect are inconsistent. One study has previously reported that miR-525-5p was increased in laryngeal squamous cell carcinoma to exert a cancer promoting effect (*Cybula et al., 2016*). However, multiple recent studies have highlighted the tumor suppressor properties of miR-525-5p. It has been reported that miR-525-5p negatively regulates glioma cell proliferation and epithelial mesenchymal transition, and this is mediated by targeting STAT1 (*Xie et al., 2020*). Other reports suggest that low

expression of miR-525-5p is closely associated with poor prognosis in thymic cancer patients (*Wang et al., 2021*). In addition, LINC01207 promoted lung cancer cell growth, while miR-525-5p acts a tumor suppressive role by sponging LINC01207 (*Zhang, Jin & Zhang, 2022*). Moreover, miR-525-5p expression was downregulated in chordoma and breast cancer progression and positively correlated with tumor size (*Zhang et al., 2021*; *Yu et al., 2022*). In this study, miR-525-5p was low expression in tissues and cells of lymphoma tumor, and miR-525-5p inhibited cell proliferation, invasion and clonogenicity, and promoted apoptosis. *In vivo* studies showed that miR-525-5p overexpression suppressed tumor growth in nude mice.

Myd88 was originally isolated from myeloid cells and consists of 296 amino acid residues and is a member of the death domain family and the Toll/interleukin-1 (IL-1) receptor (TLR) family. A study showed that Myd88 silencing significantly inhibited the growth and invasion of colorectal cancer cells SW480 and HCT116 (*Zhu et al., 2020*). Another study showed that in the sepsis related acute kidney injury cell model established by lipopolysaccharide stimulated HK-2 cells, Myd88 inhibited cell growth and promoted inflammation and oxidative stress (*Zhou, Qing & Xu, 2021*). In the progression of hepatocellular carcinoma, the activation of Myd88 signaling could promote the proliferation and migration of hepatitis B virus-infected LX-2 cells, and inhibit apoptosis (*Yuan et al., 2021*).

Moreover, Myd88 is a key target molecule in the downstream transduction of TLR pathways, as well as an important bridge to activate NF-κB pathway. The high expression of TLRs is closely related to tumor proliferation, invasion and metastasis. Key molecules in TLR4 related signaling pathways include TLR4, Myd88, TRAF6 and NF-κB (*Shao et al., 2021*; *Long et al., 2018*). After being recognized by different molecular patterns, TLR4 initiates downstream related signaling pathways, including Myd88 dependent pathway and Myd88 independent pathway. After TLR4 activates Myd88, the activated Myd88 aggregates and acts on TRAF6 to activate the downstream NF-κB signaling pathway. In response to Myd88, the NF-κB inhibitory kinase complex, composed of IκK-α, IκK-β, and IκK-γ, stimulates the phosphorylation of IκB. This action further lifts the repression of NF-κB, enabling it to move into the nucleus, which promotes gene transcription and significantly mediates gene expression. Many studies have shown that NF-κB is continuously activated in most tumors and promotes tumor initiation, progression, and metastasis. Ding et al found that inhibiting the overexpression of key genes of TLR4/Myd88/NF-κB signaling pathway can significantly reduce the incidence of diethylnitrosamine induced liver cancer (*Ding et al., 2019*). *Zhang et al. (2017)* found that inhibiting TLR4/NF-κB/MMP-9 signaling pathway could inhibit the proliferation of colon cancer cells, induce their apoptosis, and improve caspase 3/9 activity. In this study, Myd88 was upregulated in lymphoma tissues and Myd88 silencing inhibited proliferation and clonogenicity of lymphoma cells, and promoted tumor cell apoptosis, and that Myd88 reversed the regulation of miR-525-5p on tumor cells. Independently, depletion of A20 protein, the NF-κB signaling negative regulator, further enhanced Myd88 L265P mediated NF-κB activation and lymphoma growth (*Yu et al., 2021*). *Jin et al. (2020)* showed that

SPOP inhibits DLBCL cell growth *in vitro* and tumor xenografts *in vivo* by inhibiting Myd88/NF-κB signaling.

Although we found that miR-525-5p inhibited the proliferation, invasion and clonality of lymphoma cells, and promoted apoptosis, and further study found that this was mediated by inhibiting the NF-κB pathway through targeted inhibition of Myd88 expression, in this study, we detected the expression of miR-525-5p in three lymphoma cell lines, but only used a single lymphoma cell line U2932 for research, which has certain limitations. Later, we will further validate the results of this study in the remaining two lymphoma cell lines.

## CONCLUSIONS

This study confirmed that miR-525-5p inhibits the progression of DLBCL by inhibiting the MyD88/NF-κB pathway *in vivo* and *in vitro*. Although our research is still at the molecular level and a small number of animal experiments, our results fill the gaps in previous studies and may provide a new reference for the treatment of DLBCL.

## ABBREVIATIONS

| | |
|---|---|
| **DLBCL** | diffuse large B cell lymphoma |
| **miRNAs** | microRNAs |
| **TLRs** | toll like receptors |
| **Myd88** | myeloid differentiation factor 88 |
| **NF-κB** | nuclear factor kappa B |

### Funding

This work is supported by the Beijing Life Oasis Public Welfare Service Center (cphcf-2022-15). The funders had no role in study design, data collection and analysis, decision to publish, or preparation of the manuscript.

### Grant Disclosures

The following grant information was disclosed by the authors:
Beijing Life Oasis Public Welfare Service Center: cphcf-2022-15.

### Competing Interests

The authors declare that they have no competing interests.

### Author Contributions

- Xiuchen Guo conceived and designed the experiments, performed the experiments, analyzed the data, prepared figures and/or tables, authored or reviewed drafts of the article, and approved the final draft.
- Jingbo Zhang conceived and designed the experiments, analyzed the data, authored or reviewed drafts of the article, and approved the final draft.

- Jingya Zeng performed the experiments, prepared figures and/or tables, and approved the final draft.
- Yiwei Guo performed the experiments, prepared figures and/or tables, and approved the final draft.
- Lina Zhao conceived and designed the experiments, analyzed the data, authored or reviewed drafts of the article, and approved the final draft.

## Human Ethics

The following information was supplied relating to ethical approvals (*i.e.*, approving body and any reference numbers):

All samples obtained in this study were approved by the ethics committee of the Harbin Medical University Cancer Hospital and abided by the ethical guidelines of the Declaration of Helsinki.

## Animal Ethics

The following information was supplied relating to ethical approvals (*i.e.*, approving body and any reference numbers):

All protocols on animals conformed to the laboratory animal guidelines. This study was approved by the Animal Ethics Committee of Harbin Medical University Cancer Hospital.

## Data Availability

The raw data is available in the Supplemental Files.

## Supplemental Information

Supplemental information for this article can be found online at http://dx.doi.org/10.7717/peerj.16388#supplemental-information.

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
