# Peer review of "MiR-525-5p inhibits diffuse large B cell lymphoma progression via the Myd88/NF-κB signaling pathway"

_PeerJ, doi:10.7717/peerj.16388_

## Round 0.1 · original submission · Major Revisions

The flaws and weaknesses of this article are that the results section (Fig. 1, 2, 3, 4, 5, and 6) is not written in enough detail, and the author needs to make the readers understand the background of each experiment, the purpose, and the methodology used, rather than just listing the results. Also, there are many grammatical errors in this article and proper English editing is needed.

Please respond and make appropriate revisions based on the reviewers' suggestions and my comments (below). This will greatly improve the quality of the manuscript.

My comments:
1. Line 17, as well as the remaining part of this manuscript: [3' UTR] should be revised to [3'-UTR]. Please carefully check and make the according revisions.
2. Line 68: [In conclusion]? In the Introduction section, the use of [In conclusion] is inappropriate. Please replace it with other words.
3. Line 69: [gene-loss and gain assay]? This writing methodology is inaccurate. miRNAs are untranslated RNA molecules, not genes.
4. Line 86: [A total 10] should be [A total of 10]. [4-6 week] should be [4-6 weeks].
5. Line 90: [accesses] should be [access].
6. Line 102: [was used to measured]?
7. Line 104: [were transferred] should be [was transferred].
8. Line 105: [After blocked] should be [After blocking].
9. Line 133: [to analysed] should be [to analyze].
10. The conclusion section needs to highlight the scientific value and future clinical applications of this study.
11. Molecular weight has been shown in all western blotting results.
12. The extent to which this study fills a gap in previous research must be clearly shown in the Abstract and Introduction sections.

Reviewer 1 ·

Basic reporting

In this study, the author's findings suggest that miR-525-5p may be involved in the progression of diffuse large B-cell lymphoma (DLBCL). They found that miR-525-5p inhibited the proliferation, invasion, and clonality of U2932 cells, and increased cell apoptosis, while miR-525-5p silenced and enhanced the growth of tumor cells. In mechanism research, they found that miR-525-5p inhibits the NF-κB pathway by targeting the 3'UTR of Myd88, thereby delaying the progression of DLBCL inhibition. Overall, the author's research approach is clear and the experimental design is complete, but there are still some limitations that need to be revised in the manuscript to better improve its quality.

Experimental design

1. In Figure 1C, the author detected the expression levels of miR-525-5p in multiple lymphoma cell lines and selected one cell line for subsequent research. Have other cell lines been studied accordingly? I think the author should be discussing this limitation.

Validity of the findings

1. I noticed that the author used Myd88 siRNA to knock down its expression level, but I did not see the sequence information of siRNA in the manuscript. Please supplement this information.
2. Similarly, the manuscript displays primer information for RT-qPCR, but the primer sequence information for Myd88 and its reference gene is missing, as this method was used to detect mRNA expression of Myd88 in Figure 3C.
3. In Figure 3E, the expression of MyD88 in the tumor tissue of patient 2 in the Western blot band appeared to be inconsistent with the statistical map, please check and clarify.

Additional comments

1. Although the current legend contains basic information, it is too short and lacks some important details, including experimental grouping, transfection time, and other information.

Reviewer 2 ·

Basic reporting

Diffuse large B-cell lymphoma (DLBCL) is a highly invasive B-cell lymphoma. In this study, the author investigated the regulatory mechanism of miR-525-5p in tumor progression through in vitro cell models and in vivo tumor bearing mouse models. In vitro, they altered miR-525-5p in the lymphoma cell line U2932 by overexpressing plasmids and small interfering RNA, and found that miR-525-5p had tumor inhibitory effects. Next, they identified the downstream target Myd88 of miR-525-5p for further research and found that Myd88 promotes U2932 cell growth, which is mediated by activating the NF-κB signaling pathway. In the in vivo model, the author validated the inhibitory ability of miR-525-5p on tumor formation. Overall, this study is very interesting, and I think it would be better if more details could be explained about the experimental steps.

Experimental design

a. I missed the “n”, the number of samples used in the experiments. This number should be mentioned in legends of every figure. For the animal experiments it was not mentioned how many rats were used for every experimental group, nor the numbers used for histology, Western Blot, cytokine measurements etcetera.
b. Explain the principle of flow cytometry for detecting cell apoptosis, that is, how to define the horizontal and vertical coordinates. Provide the information how flow cytometry data was evaluated. Provide the information about flow cytometer manufacturer.
c. The detection samples in Figures 1A and 6D&6E are tumor tissue. Please provide relevant data on tissue sample fixation and processing.
d. In Figure 1A, the author detected the expression of miR-525-5p in adjacent cancer tissues and tumor tissues. How to define adjacent cancer tissues? How was it obtained?
e. In the materials and methods section, the author wrote " The cells were incubated in RPMI 1640 medium (10% FBS, 100 U/mL penicillin, 100 μg/mL streptomycin). " As far as I know, DMEM is often used in the culture medium of tumor cells,please, explain why U2932 cells were grown in RPMI instead of DMEM as ATCC recommends.
f. Throughout the study- cells were transfected and not the various construct was transfected into cells (lines 117-118).

Validity of the findings

g. The authors proposed Myd88 as a possible target of miR-525-5p. Are there any previous reports or data that demonstrated the contribution of Myd88 to proliferation, invasion and apoptosis?

Additional comments

h. The downstream target of miR-525-5p is not just Myd88. How do you find it?
i. Some typing errors should be checked throughout the manuscript. A careful grammar and spelling check should be performed. Please revise the results section and polish the language.

Reviewer 3 ·

Basic reporting

The manuscript entitled “MiR-525-5p inhibits diffuse large B cell lymphoma progression via the Myd88 / NF-κB signaling pathway” aimed to clarify the effects of miR-525-5p and its target gene MyD88 on proliferation, invasion, clonogenesis and apoptosis of lymphoma cells through in vivo and in vitro studies, and to explore the mechanism of miR-525-5p inhibiting lymphoma progression, aiming to provide a new reference strategy for targeted gene therapy of lymphoma. Although this is an interesting study, I still have some concerns about the article. As follows.

Experimental design

1. In 2.3 CCK-8 assay, the cell density of 96 well plate should be checked.
2. Figure 1C shows the expression of miR-525-5p in various lymphatic cancer cell lines. What about the expression of Myd88? Is it necessary to detect the expression of Myd88 in various lymphatic cancer cell lines?
3. For Western blot analysis, antibody information is too simple. Species origin? Polyclonal antibody or monoclonal antibody? What secondary antibodies were used? How to visualize protein bands? How to quantify protein expression? None of this information is missing.
4. How does Myd88 affect NF-κB expression? In Figure 5, the results show that overexpression of Myd88 once again promotes the protein expression of NF-κB, thereby affecting changes in cell behavior. What is the possible mechanism behind this? It should be mentioned in the discussion.

Validity of the findings

1. Add a molecular weight marker to all western blot images in this manuscript.
2. More thorough explanations and important details on the experimental procedures is missing throughout the paper.

Additional comments

1. The introduction is not well written, for that it gives too few descriptions about MyD88. It is necessary to add more literature to clarify whether Myd88 plays a tumor suppressor or tumor promoting role in a variety of tumors.

---

## Round 0.2 · Minor Revisions

Issues that need to be further revised or improved:
1. Line 20: [mimetics] should be changed to [mimics].
2. Line 1, line 23, line 26, and line 258: [Myd88 / NF-κB] should be revised to [Myd88/NF-κB] so that extra spaces can be removed.
3. Line 242: [TLR4 / Myd88 / NF-κB] should be revised to [TLR4/Myd88/NF-κB] so that extra spaces can be removed.
4. Line 37-38: [With the continuous innovation … from the genetic level]: Necessary citations should be added.
5. Line 133: [the luciferase reporter vector was fused into the wild-type and Mutant sequences of Myd88 respectively] should be revised to [the wild-type and mutant Myd88 3’-UTR sequences were fused into the luciferase reporter vector respectively].
6. Line 267: [and miR-525-5p inhibitor reversed the inhibitory effect of miR-252-5p mimic] This expression is inaccurate because there is no rescue experiment being used here. Therefore, this sentence should be changed to [The silencing of miR-525-5p with miR-525-5p inhibitor promoted these malignant features of U2932 cells (Fig. 2A-D)].
7. Line 173: [wild type and the mutant sequence of Myd88 were] should be revised to [wild type and the mutant sequences of Myd88 3’-UTR were].
8. Fig. 5C and 5D: It seems that a bar graph of the percentage of apoptosis was placed next to the images of the cell invasion experiment. A bar graph of the percentage of cell invasion should be shown here. Please change the location of the bar graphs in Fig. 5C and 5D accordingly.
9. Line 217 and 218: [lncRNA 01207] should be revised to [LINC01207].
10. Line 224: [Toll / interleukin 1 receptor family] should be revised to [Toll/interleukin-1 (IL-1) receptor (TIR) family].
11. Line 233 and 234: [Key molecules in TLR4 related signaling pathways include TLR4, Myd88, TRAF6 and NF-κB]: Please cite the relevant reference.
12. Line 237-240: [In response to Myd88, the NF-κB inhibitory kinase complex formed by IκK-α, IκK-β and IκK-γ will promote IκB phosphorylation, further de repressing NF-κB, allowing NF-κB to translocate into the nucleus, promoting gene transcription and allowing gene expression to rise significantly]: This part should be re-written.
Here is an example: In response to Myd88, the NF-κB inhibitory kinase complex, composed of IκK-α, IκK-β, and IκK-γ, stimulates the phosphorylation of IκB. This action further lifts the repression of NF-κB, enabling it to move into the nucleus, which promotes gene transcription and significantly mediates gene expression.

Reviewer 1 ·

Basic reporting

The author has made good standardization and revisions based on my review comments, and I have no further comments.

Experimental design

Nice modification. I have no further comments

Validity of the findings

Nice modification. I have no further comments

Additional comments

The author made point by point revisions to my comments and provided serious responses, greatly enhancing the quality of this manuscript after the revisions. I think this article is sufficient to meet the publication standards of the magazine.

Reviewer 2 ·

Basic reporting

After modification, the structure of the article is clear and clear, and professional English is used throughout the entire process. References provide sufficient background.

Experimental design

After revision, the research questions in the article have become clearly defined, relevant, and meaningful. The described method already has sufficient details and information for replication.

Validity of the findings

The data provided by the author is robust, statistically reliable, and controllable. And the conclusion section was fully stated and well revised.

Additional comments

I have no further opinions and believe that the article can be published.

Reviewer 3 ·

Basic reporting

The basic structure of the article is clear and the language is professional.

Experimental design

The content of the experimental design section has been modified and improved by the author, and the method section has been described appropriately.

Validity of the findings

The description in the results section has also been well optimized, and the provided data is reasonable.

Additional comments

The article has met the requirements for magazine publication and is believed to be able to be published.

---

## Round 0.3 · accepted · Accept

The authors have addressed all of my comments. I think this revised version could be considered for publication in PeerJ.